

# A possibility of large scale intrusions generation in the Arctic Ocean under stable-stable stratification: an analytical consideration

Natalia Kuzmina

Shirshov Institute of Oceanology, 36 Nakhimovsky Ave, 117997 Moscow, Russia

*Correspondence to:* Natalia Kuzmina (kuzmina@ocean.ru)

**Abstract.** Some analytical solutions are found for the problem of three-dimensional instability of a weak geostrophic flow with linear velocity shear taking into account vertical diffusion of buoyancy. The analysis is based on the potential vorticity equation in a long-wave approximation when the horizontal scale of disturbances is taken to be much larger than the local baroclinic

Rossby radius. It is hypothesized that the solutions found can be applied to describe stable and unstable disturbances on a planetary scale with respect, especially, to the Arctic Basin, where weak baroclinic fronts with typical temporal variability periods of the order of several years or more are observed and the beta-effect is negligible. Stable (decaying with time) solutions describe disturbances that, in contrast to the Rossby waves, can propagate to both the west and east, depending on the sign of the linear shear of geostrophic velocity. The unstable (growing with time) solutions are applied to describe large-scale intrusions at

baroclinic fronts under the stable–stable thermohaline stratification observed in the upper layer of the Polar Deep Water in the Eurasian Basin. The proposed description of intrusive layering can be considered as a possible alternative to the mechanism of interleaving due to the differential mixing.

## 1 Introduction

Study of the intrusions in oceanic frontal zones is necessary to analyse the mechanisms of ventilation and mixing in the ocean

interior (see, for example, Zhurbas et al., 1983, 1987; Rudels et al., 1999, 2009; Kuzmina and Zhurbas, 2000; Walsh and Ruddick, 2000; Merryfield, 2000; Radko, 2003; Richards and Edwards, 2003; Kuzmina et al., 2005, 2011; Smyth and Ruddick, 2010). Intrusive layering, as a rule, results from the instability of oceanic fronts. One of the major mechanisms responsible for the instability of both thermohaline and baroclinic fronts is related to double diffusion (Stern, 1967; Ruddick and Turner, 1979; Toole and Georgi, 1981; McDougall, 1985a, 1985b; Niino, 1986; Yoshida et al., 1989; Richards, 1991; Kuzmina and Rodionov,

1992; May and Kelley, 1997; Kuzmina, 2000). However, in the Eurasian Basin of the Arctic Ocean there are baroclinic and thermohaline fronts within the upper layer of the Polar Deep Water (PDW) populated with intrusive layers of vertical length scale as large as 30 m and horizontal scale reaching more than 100 km (Rudels et al., 1999, 2009; Kuzmina et al., 2011) observed at stable–stable stratification (i.e., when the mean salinity increases with depth while the mean temperature decreases with depth). It can be suggested that thermohaline intrusions within the upper layer of PDW are driven by differential mixing.

Merryfield (2002) was the first to show satisfactory agreement between calculations of unstable modes from a 3D interleaving model, taking into account differential mixing at a no baroclinicity front and observations of intrusive layering at a pure thermohaline front in the PDW. Merryfield's (2002) findings were confirmed by Kuzmina et al. (2014). However, the 2D model of interleaving driven by differential mixing at the baroclinic front did not show a satisfactory fit simultaneously between the three modelled parameters, namely the vertical scale, growth time and slope of the fastest growing mode, and observations of

intrusions in a frontal zone with substantial baroclinicity in the upper PDW layer (Kuzmina et al., 2014). In particular, it was found that the vertical scale of the most unstable mode is two to three times smaller than the vertical scale of intrusions observed





in the baroclinic front. Furthermore, it is worth noting that the 2D models of double-diffusive interleaving, as applied to typical baroclinic fronts in the ocean, are able to forecast intrusive layers of no more than 10 m vertical length scale (Kuzmina and Rodionov, 1992; May and Kelley, 1997, 2001; Kuzmina and Zhurbas, 2000; Kuzmina and Lee, 2005; Kuzmina et al., 2005). Therefore, despite the fact that there are proven-by-simulation hypotheses of a merger of small vertical-scale intrusions into larger structures (Radko, 2007), new approaches to the mathematical description of the formation of large intrusions in areas of baroclinic fronts appear relevant.

We suggest that interleaving at baroclinic fronts may be considered as a result of 3D instability of weak geostrophic current due to the combined effects of vertical shear and diffusion of density (buoyancy).

The effect of vertical diffusion of buoyancy on the baroclinic instability of geostrophic zonal wind was studied theoretically by Miles (Miles, 1965). Based on an analogy between the equations describing the dynamics of large-scale atmospheric perturbations and the Orr–Sommerfeld equation (Lin, 1955), Miles analysed the instability of the critical layer (the very thin layer in which the phase velocity of disturbance is equal to the velocity of zonal flow). As a result, Miles built an analytical asymptotic solution taking into account the very small, but finite vertical diffusion of buoyancy. Based on the analysis, Miles concluded that the influence of vertical diffusion of buoyancy in destabilising the zonal wind is not essential in comparison with baroclinic instability (the generation of cyclones and anticyclones) for typical atmospheric geostrophic wind. One can assume, however, that other situations can be observed in the deep ocean. Indeed, in the Polar zones, for example, in the Eurasian Basin of the Arctic, very weak geostrophic currents are observed at deep layers (Aagaard, 1981). These currents can have a large horizontal (transverse) scale and large time scale of variability, the latter being estimated at much more than one year (Aagaard, 1981). Taking into account that the influence of the $\beta$-effect on the dynamics of large-scale disturbances is negligible in the Polar Ocean, it seems reasonable to suggest that the role of diffusion of buoyancy in the destabilization of weak geostrophic currents can be important. Therefore, in such circumstances one would expect the formation of intrusions, rather than vortices.

The present work is devoted to the search for analytical unstable (increasing with time) and stable (decreasing with time) solutions based on the potential vorticity equation describing the 3D dynamics of a weak baroclinic front, with the vertical diffusion of buoyancy included. The results, hopefully, will make it possible to obtain some new conceptions about the causes of the formation of large intrusions, particularly in the regions of the Arctic Basin with stable–stable stratification.

## 2 Problem formulation, derivation of basic equation, and solution search

Let us consider the problem of the 3D instability of the baroclinic front on the basis of the linearized equations of motion in quasi-geostrophic approximation (see, for example, Pedlosky, 1979; Cushman-Roisin, 1994):

$$U = -\frac{1}{f}\frac{\partial P}{\partial y}, \ V = 0, \ W = 0, \ \frac{\partial P}{\partial z} = -g\overline{\rho} \tag{1}$$

$$\frac{\partial p}{\partial z} = -g\rho, \ u = -\frac{1}{f}\frac{\partial p}{\partial y}, \ v = \frac{1}{f}\frac{\partial p}{\partial x} \tag{2}$$

$$\left(\frac{\partial}{\partial t} + U\frac{\partial}{\partial x}\right)\left(\frac{1}{f}\Delta p\right) + \beta v - f\frac{\partial w}{\partial z} = \frac{\tilde{K}}{f}\frac{\partial^2}{\partial z^2}\Delta p \tag{3}$$



$$\left(\frac{\partial}{\partial t}+U\frac{\partial}{\partial x}\right)\rho + v\frac{\partial\overline{\rho}}{\partial y} - \frac{N^2}{g}w = K\frac{\partial^2}{\partial z^2}\rho, \tag{4}$$

where $U$ and $V$ are zonal and meridional components of the geostrophic velocity, $P$ and $\overline{\rho}$ are the mean pressure and density both divided by the reference density, $N, f$ and $g$ are the buoyancy frequency, Coriolis parameter and gravity acceleration, $u, v$ and $w$ are velocity fluctuations along the $x, y$ and $z$ axes, respectively, $p$ and $\rho$ are pressure and density fluctuations both

divided by the reference density, $\beta = \partial f / \partial y$, $\Delta = \partial^2 / \partial x^2 + \partial^2 / \partial y^2$, and the $x, y$ and $z$ axes are directed eastward, northward and upward, respectively. Vertical friction with a constant coefficient $\tilde{K}$ is considered in the vorticity Eq. (3). The density advection Eq. (4) accounts for only the vertical diffusion with a constant coefficient $K$. The constant coefficients $\tilde{K}$ and $K$ are treated as average values in an ocean layer under investigation.

Let us take the distribution of mean density, normalized to the reference density, as follows:

$$\overline{\rho}(z,y) = fsyz/g + f\tilde{s}y/g - \frac{N_0^2}{g}z + 1, \tag{5}$$

where $N_0 = const > 0$ is a characteristic value of the buoyancy frequency in the frontal zone, and $\tilde{s}$ and $s$ are dimensional constants, either positive or negative, that characterize the cross-front gradients of density and the vertical shear of the basic geostrophic flow.

The first term on the right of Eq. (5) has not been taken into account in the interleaving models describing both the 2D (see,

for example, Kuzmina, Rodionov, 1992; May, Kelley, 1997; Kuzmina, Zhurbas, 2000) and 3D (Eady, 1949; Miles, 1965; Smyth, 2008) instabilities of the oceanic baroclinic fronts. However, the oceanic fronts can be characterized by not only the cross-front gradient of density, but also the cross-front gradient of the buoyancy frequency. This is the case described by Eq. (5): the squared buoyancy frequency, $N^2 = -gd\overline{\rho}/dz$, is a linear function of $y$. This dependence is assumed to be weak: $|s|fL << N_0^2$, where $L$ is the characteristic lateral length scale (width) of the frontal zone ($0 \le y \le L$). However, even a weak lateral change in the

buoyancy frequency indicates the existence of a quadratic dependence of geostrophic velocity on the vertical coordinate $z$. Indeed, if the mean density distribution is expressed by Eq. (5), the geostrophic current velocity will be

$$U = U_1 + U_2 + U_3, \ U = sz^2/2, \ U_2 = \tilde{s}z, \ U_3 = const, \tag{6}$$

where $U_1$, $U_2$ are the constituents of geostrophic velocity with linear ($U_1$) and constant ($U_2$) vertical shear: $dU_1/dz = sz$, $dU_2/dz = \tilde{s}$; $U_3$ is the barotropic (constant) velocity addition.

The equation of evolution of potential vorticity, derived on the basis of Eqs. (1–4) is

$$\left(\frac{\partial}{\partial t}+U\frac{\partial}{\partial x}\right)\left(\frac{\partial^2 p}{\partial z^2}+\frac{N_0^2\Delta p}{f^2}\right)+\frac{\beta v N_0^2}{f}-vsf = K\frac{\partial^4}{\partial z^4}p+\tilde{K}\frac{N_0^2}{f^2}\frac{\partial^2}{\partial z^2}\Delta p \ . \tag{7}$$

Equation (7) was derived under the abovementioned assumption of $|fsL/N_0^2| << 1$. Note that in the differentiation of Eq. (4) with respect to $z$, such members as $(\partial U/\partial z)\cdot(\partial\rho/\partial x)$ and $(\partial v/\partial z)\cdot(\partial\overline{\rho}/\partial y)$ are reduced, since they are equal in magnitude and opposite in sign, in accordance with Eqs. (1) and (2).





As can be seen from Eq. (7), the last term on the left can strengthen or weaken, depending on the sign of $s$, the impact of the $\beta$-effect on the dynamics of disturbances.

We will consider at $K \approx \widetilde{K}$ long-wave disturbances (perturbation on a planetary scale) of weak geostrophic current ($F(z) = B_1 F_1(z) + B_2 F_2(z) + B_3 F_3(z)$) which satisfy the following relationship between the vertical and horizontal length scales

($H$ and $\widetilde{L}$, respectively): $\widetilde{L} \gg L_R$, where $L_R = N_0 H / f$ is the baroclinic Rossby radius of deformation. If we apply Eqs. (1–4) to describe the motion in the Arctic Basin, the $\beta$-effect term can be ignored because $\beta \approx 0$ in the vicinity of the North Pole.

Taking into account the abovementioned conditions, we may use the method of series expansion at small parameter $\delta^2 = N_0^2 H^2 /(\widetilde{L}^2 f^2) = Bu$, where $Bu$ is the Burger number (see, for example, Cushman-Roisin, 1994). At $\delta^2 \sim 10^{-3} - 10^{-4}$ ($\widetilde{L} \sim 10^4 - 10^5 \,\text{m}$, $20 < H < 100$ m, $N_0 \sim 10^{-3}\,\text{s}^{-1}$, $f \sim 10^{-4}\,\text{s}^{-1}$), it is reasonable to consider only the first term

of the series. In this case, we can rewrite the potential vorticity equation in the simplified form:

$$\left(\frac{\partial}{\partial t} + U \frac{\partial}{\partial x}\right)\left(\frac{\partial^2 p}{\partial z^2}\right) - vsf = K \frac{\partial^4}{\partial z^4} p \,. \tag{8}$$

The relative error committed in the solution by so doing is expected to be of the order of $\delta^2$, and the smaller $\delta^2$, the smaller the error.

In accordance with our approximation, Eq. (8) corresponds to the advection equation (4) at $w = w_0 = const$:

$$\left(\frac{\partial}{\partial t} + U \frac{\partial}{\partial x}\right)\rho + v \frac{\partial \overline{\rho}}{\partial y} - \frac{N_0^2}{g} w_0 = K \frac{\partial^2}{\partial z^2} \rho \,. \tag{9}$$

Thus, the vorticity equation (3) drops out of consideration. Indeed, given that the diffusivity of mass, $K$, in the oceanic interior (particularly in the deep water of the Arctic Ocean) probably does not exceed the value of $1 \times 10^{-5}\,\text{m}^2\text{s}^{-1}$ and the vertical length scale of the intrusions, $H$, to which this theory is applied is approximately equal to $H \sim 20 \div 100$ m, the ratio of $U/\widetilde{L}$ is estimated as $U/\widetilde{L} \leq 10^{-8}\,\text{s}^{-1}$. Based on the latter estimate, one can suggest that the vertical circulation caused by the temporal

change of vorticity and frictional force will not significantly affect the dynamics of large-scale disturbances. This hypothesis will be tested a posteriori by analysing the solutions obtained.

Given that $w_0 = const$, we can put $w_0 = 0$ in Eq. (9), and therefore rewrite Eq. (9) as

$$\left(\frac{\partial}{\partial t} + U \frac{\partial}{\partial x}\right)\rho + v \frac{\partial \overline{\rho}}{\partial y} = K \frac{\partial^2}{\partial z^2} \rho \,. \tag{10}$$

Based on the above, we can conclude that the slow extra-large-scale disturbances of weak geostrophic flow are described by

the quasi-stationary system of Eqs. (2) and (10).

Let us now pay attention to an important issue. Namely, if we suppose that $U(z) = 0$ in Eq. (10) and consider salt fingering instead of diffusion of buoyancy, then, in addition to Eq. (2), it will be necessary to write the following two equations instead of Eq. (10):

$$\frac{\partial \rho}{\partial t} = K_s (1-\gamma)\widetilde{\beta} \frac{\partial^2 S}{\partial z^2} \,, \quad \frac{\partial S}{\partial t} + v \frac{\partial \overline{s}}{\partial y} = K_s \frac{\partial^2 S}{\partial z^2} \,, \tag{10'}$$





where $S$ and $\bar{S}$ are the salinity disturbance and mean, $K_S$ and $\gamma = \tilde{\alpha}F_T / \tilde{\beta}F_S < 1$ are the vertical diffusivity of salinity and the flux ratio for salt finger convection, $F_T$ and $F_T$ are the vertical fluxes of temperature and salinity, $\tilde{\alpha}$ and $\tilde{\beta}$ are the temperature expansion and salinity contraction coefficients, respectively.

Equations (10´) along with (2) constitute the system of equations that was used by Stern (1967) to obtain the polynomial

dependence between the growth rate of unstable perturbations, wave numbers and hydrological parameters [see Eq. (4) of (Stern, 1967)]. Therefore, the proposed model, which consists of Eqs. (2) and (10), can in a certain sense be regarded as an analogue of the model by Stern (1967) for investigating the interleaving on a large horizontal scale. However, the derivation of the model equations, which we have done above, is useful to understand the limits of the model's applicability.

From the point of view of the author of this work, a simple quasi-stationary (geostrophic) system of equations accurately

describes the large scale movement especially in the Arctic Ocean, where the influence of the beta effect is not significant, the baroclinic fronts of large width in the ocean interior are often not intense (Kuzmina et al., 2011), and the baroclinic radius of deformation, $HN/f$, at $H \sim 100$ m, does not exceed 2–5 km (see also Section 3).

To analyse the instability of the geostrophic flow, let us consider a layer with a vertical scale of $2H_0$ and place the co-ordinate system on the middle line of the layer. For the analysis of the instability in the frame of Eqs. (2) and (10), we will

consider a symmetric relative to the midline of the layer geostrophic flow with quadratic $z$-dependence of velocity:

$$U = U_1 + U_3 = sz^2/2 + U_3.$$

A parabolic dependence of the geostrophic flow velocity upon the vertical co-ordinate can be observed in the rotary flow of the intra-pycnocline vortices, as well as in many other ocean flows. In any case, as mentioned above, in the oceanic frontal zones it is not unlikely to observe changes of the buoyancy frequency in the cross-front direction, which indicate the existence of linear

shear of geostrophic velocity. Consideration of geostrophic flow instability with the velocity profile of $U = U_1 + U_2 + U_3$ is also possible on the basis of analytical methods, and we will consider the related issues below.

Let us discuss the conditions on the boundaries of the layer in relation to the ocean. Keeping in mind the Eady problem (Eady, 1949), one has to require the vanishing of vertical velocity at the layer boundaries. In accordance with our approximation, this condition is satisfied.

Due to the fact that the model takes into account vertical diffusion, it is logical to take the conditions of zero buoyancy flux (for density perturbations) at the layer boundaries: $p_{zz} = 0$ at $z = \pm H_0$ (the type 1 boundary conditions). It is reasonable to consider another type of condition too, namely, the slippery boundary conditions or equivalent conditions of the vanishing density disturbances at the boundaries: $dv/dz = du/dz = \rho = 0$ at $z = \pm H_0$ (the type 2 boundary conditions). Under the type 2 boundary conditions, it is necessary to require the absence of convergence or divergence of buoyancy flux within the layer:

$p_{zz}(z = H_0) = p_{zz}(z = -H_0)$. This requirement is necessary because the convergence or divergence of the buoyancy flux within the layer may increase or, conversely, decrease the stability of the layer.

Using (2), we rewrite (10) as:

$$\left( \frac{\partial}{\partial t} + U \frac{\partial}{\partial x} \right) \left( \frac{\partial p}{\partial z} \right) - \frac{\partial p}{\partial x} sz = K \frac{\partial^3}{\partial z^3} p, \tag{11}$$

where $U = U_1 + U_3$.





To analyse the instability of geostrophic flow, we will seek the solution of Eq. (11) at $L = \tilde{L}$ in the form

$$p = \mathrm{Re}\{F(z)e^{ik(x-ct)}\sin(\pi y/L)\}, \tag{12}$$

where $k$ is the wave number along the $x$ axis and $c$ is the growth rate. Disturbances of the horizontal velocities will be expressed as $u = -p\pi/Lf$ and $v = pik/f$. The solution will be unstable, i.e., increasing with time, if the imaginary part of $c$ is positive:

$\mathrm{Im}(c) > 0$.

Substituting (12) into (11) yields the following equation:

$$ik(U_1 + U_3 - c)\left(\frac{dF(z)}{dz}\right) - F(z)iksz - K\frac{d^3}{dz^3}F(z) = 0. \tag{13}$$

We are interested in finding an answer to the following question: is it possible to make certain judgements about the possibility of instability of geostrophic flow in a finite vertical layer, based on the analytical solutions of Eq. (13) at some values

of parameter $c$?

It is easy to verify that the following functions are the partial solutions of Eq. (13):

$$F_1(z) = e^{-az^2/2}, \quad F_2(z) = az^2 + D, \tag{14}$$

where $a^2 = iks/2K$, $D = -2a(c - U_3)/s$, $ikc = ik(c_1 + ic_2) = 5a \cdot K + ikU_3$, $c_1 = \mathrm{Re}\,c$, $c_2 = \mathrm{Im}\,c$, and $U_1 + U_3 - \mathrm{Re}\,c \neq 0$ for any point $z = z_0$ lying inside the layer.

To test partial solutions (14), one has to substitute $F_1(z)$ and $F_2(z)$ from (14) into Eq. (13), reduce the latter to a cubic polynomial $P(z) = A_3 z^3 + A_2 z^2 + A_1 z^1 + A_0 z^0$ and evaluate the coefficients $A_0$, $A_1, A_2$, and $A_3$. It is easy to obtain that this polynomial is identically zero (i.e., $A_0 \equiv 0$, $A_1 \equiv 0$, $A_2 \equiv 0$, and $A_3 \equiv 0$).

Based on the theory of ordinary differential equations (see, for example, Polyanin and Zaitsev, 2001) and taking into account that the functions $F_1(z)$ and $F_2(z)$ are linearly independent, we can write the general solution of Eq. (13) for $ikc = 5a \cdot K + ikU_3$

in the form:

$$
\begin{aligned}
&F(z) = B_1 F_1(z) + B_2 F_2(z) + B_3 F_3(z) \\
&F_3(z) = F_1(z) \cdot \int F_2(z)\varphi(z)dz - F_2(z) \cdot \int F_1(z)\varphi(z)dz, \\
&\varphi(z) = (F_1(z) \cdot dF_2(z)/dz - F_2(z) \cdot dF_1(z)/dz)^{-2}
\end{aligned}
\tag{15}
$$

where $B_1$, $B_2$, $B_3$ are arbitrary constants. It is important to note two facts. First, the functions $F_1(z)$ and $F_2(z)$ are even functions, while $F_3(z)$ is an odd function. Second, despite the singularity at $z = 0$ in the integrands of (15), the function $F_3(z)$ is differentiable at this point. (The last can be seen by analysing the behaviour of function $F_3(z)$ when $z \to 0$.)

Let us now consider the unstable and stable solutions of (15).

### 2.1 Unstable solutions

Solution (15) can be unstable for both $s > 0$ and $s < 0$. The real and imaginary parts of $kc$ for solutions growing with time (unstable) are



$$kc_1 = 2.5 \cdot \sqrt{|s|kK} + kU_3, \ kc_2 = 2.5 \cdot \sqrt{|s|kK} \ \text{at} \ s < 0,$$ (16 a)

$$kc_1 = -2.5\sqrt{skK} + kU_3, \ kc_2 = 2.5 \cdot \sqrt{skK} \ \text{at} \ s > 0.$$ (16 b)

Formulas (16) demonstrate that the condition $U_1 + U_3 - \text{Re}\,c \neq 0$ is satisfied for $z \in (-\infty, +\infty)$.

According to (16), the growth rate increases with the increase in $s$ and $K$, which implies that not only double diffusion, but the diffusion of buoyancy can cause instability of the geostrophic flow. However, the unstable solution is realized at $\text{Re}\,a < 0$, and hence, for any finite wave number $k$, the function $F_1(z)$ and all its derivatives increase dramatically when $z \to \pm\infty$. On the other hand, the function $F_3(z)$ and all its derivatives decrease when $z \to \pm\infty$. (This can be seen by analysing the behaviour of integrals defining function $F_3(z)$ when $z \to \pm\infty$.) Therefore, to prove the instability in a finite layer, it is necessary to show that $F(z)$ at $\text{Re}\,a < 0$ is an eigenfunction of the proper value problem with the boundary conditions of type 1 or 2 introduced above. To construct physically correct solutions we will consider two cases. Case 1, when the vertical scale of the layer corresponds to our approximation: $2H_0 \sim H \ll fL/N$. Case 2, when the vertical scale of the layer significantly exceeds the vertical scales of the disturbances for which our approximation holds true: $H \ll 2H_0$.

To satisfy the boundary conditions (either type 1 or type 2) in case 1, we have to take $B_3 = 0$, because $F_3(z)$ is an odd function. The type 1 boundary conditions are reduced to the following conditions for $F(z)$: $F_{zz} = 0$ at $z = \pm H_0$. Thus, the following equality should be satisfied:

$$e^{-aH_0^2/2}(-1 + aH_0^2) + 2B_2/B_1 = 0.$$ (17)

Given that $2B_2/B_1$ can have different values, the instability in the framework of (15) does exist, because in a wide range of typical ocean values of $H_0$, $s$, and $K$, there is the wave number $k_0 \ll f(2N_0H_0)$ at which (17) is satisfied.

The type 2 boundary conditions are reduced to $F_z = 0$ at $z = \pm H_0$. Under such conditions, the requirement of the absence of the buoyancy flux convergence/divergence within the layer is satisfied: in the case of a flow that is symmetric relative to the midline of the layer and the parity of function (15), the values of buoyancy flux at the boundaries are of the same magnitude and direction (sign). Under the type 2 boundary conditions the following equality should be satisfied:

$$e^{-aH_0^2/2} - 2B_2/B_1 = 0.$$ (18)

Obviously, in this case, as in the case of (17), there is a wave number $k_0 \ll f/(2N_0H_0)$ at which (18) is satisfied.

For case 1, graphic images of the unstable solutions corresponding to the disturbances of density $\text{Re}\,\rho = \text{Re}(dF/dz) = \tilde{\rho}$ for different boundary conditions are presented in Fig. 1. When building the solutions, typical values of hydrological parameters in relation to the Arctic Basin were used (see Subsection 2.3 and Section 3).

In case 2, we have to take $B_1 = 0$, $B_2 = 0$ and consider $F_3(z)$ as the solution of the eigenvalue problem. Indeed, $F_3(z)$ and all its derivatives sharply decrease when $z \to \pm\infty$, and, consequently, on the boundaries of the large vertical scale layer the function $F_3(z)$ and all its derivatives should be infinitesimally small, that is, the boundary conditions of type 1 and 2 are satisfied. In this case, it is relatively simply to construct an analytical solution for a more complicated form of geostrophic



current such as $U = U_1 + U_2 + U_3$. To make it, we have to seek solutions of (13) (preliminarily rewriting this equation for $U = U_1 + U_2 + U_3$) in the form:

$$\tilde{F}_1(z) = \exp(-az_*^2/2), \ \tilde{F}_2(z) = az_*^2 + D, \ z_* = z + \tilde{s}/s, \ D = -2a(c - U_3 + \tilde{s}^2/2s)/s.$$

The function $\tilde{F}_3(z)$ is constructed similarly to previous considerations (see formula (15)). This function will have an additional oscillating component in comparison with function $F_3(z)$.

For case 2, a plot of the unstable solutions corresponding to the disturbances of density $\tilde{\rho} = \mathrm{Re}(dF/dz)$ is presented in Fig. 2. It is worth noting that the function $\rho = dF_3/dz$ is differentiable at $z = 0$ as well as the function $F_3(z)$.

Thus, if the vertical diffusion of buoyancy plays a role in the dynamics of ocean processes, the long-wave perturbations of the weak baroclinic front with linear shear can be unstable (time-increasing). Note that in the case of no baroclinicity, instability due to the diffusion of buoyancy cannot arise. Indeed, when $U = 0$, Eq. (13) becomes the diffusion equation, and its general solution for the density perturbation (12) at $c \neq 0$ has the form (see, for example, Niino, 1986):

$$F'_z = \varphi(z) = Pe^{nz} + Qe^{mz},$$

where $P$ and $Q$ are integration constants, while $n$ and $m$ are expressed as $n^2 = m^2 = (-ikc/K)$, $n = +(-ikc/K)^{0.5}$, $m = -(-ikc/K)^{0.5}$.

With regard to the boundary conditions, the problem is reduced to an eigenvalue problem. Here, we briefly discuss only the evaluation of the growth rate $c_2$. It can be easily shown that the solution $\varphi(z)$ will satisfy the abovementioned boundary conditions only if $e^{4nH_0} = 1$. This equality can be satisfied if $n$ is zero or an imaginary number. The imaginary value of $n$ indicates that the value of the growth rate $c_2$ is negative for all values of the wave number $k$. In this case, the disturbance is described by trigonometric functions that decrease with time.

## 2.2 Stable solutions

Stable solutions of Eq. (13) are realized at $\mathrm{Re}\, a > 0$. In this case $F_1(z)$ and all its derivatives vanish at $z \to \pm\infty$, but $F_3(z)$ and all its derivatives increase at $z \to \pm\infty$. To construct own functions of the eigenvalue problem for the case 1 ($2H_0 \sim H \ll fL/N$), we have to take $B_3 = 0$.

The solutions describe slow time-decaying, long waves that can move, in contrast to the Rossby waves, not only to the west but also to the east according to the sign of $s$ (see Eq. (7)). Moreover, if $|s| > \beta N_0^2/f^2$ (which is quite possible especially in polar regions), the long-wave dynamics in the $\beta$-plane approximation is determined by the linear shear of geostrophic flow rather than the $\beta$-effect.

The real and imaginary parts of the growth rate of stable perturbations are

$$kc_1 = -2.5 \cdot \sqrt{|s|kK} + kU_3, \ kc_2 = -2.5 \cdot \sqrt{|s|kK} \ \text{ at } \ s < 0, \tag{19 a}$$

$$kc_1 = 2.5\sqrt{skK} + kU_3 \ \ kc_2 = -2.5 \cdot \sqrt{skK} \ \text{ at } \ s > 0. \tag{19 b}$$





In accordance with (19), the condition $U_1 + U_3 - \mathrm{Re}\,c \neq 0$ is satisfied if $2.5\sqrt{s|kK}/k > |s|H_0^{\,2}/2$. Comparing (16) and (19), we can conclude that the phase velocity of the stable and unstable disturbances has a different sign. That is, stable and unstable perturbations described by solutions (15) will move in different directions with respect to the flow and a fixed observer.

For case 1and type 2 boundary conditions, a plot of the stable solutions corresponding to the disturbances of density $\mathrm{Re}\,\rho = \mathrm{Re}(dF/dz) = \tilde{\rho}$ is presented in Fig. 3.

### 2.3 Obtained solutions: some comments

Our own functions, obtained in the previous subsections, have the vertical structure of the unstable perturbations, which differs significantly from those of the classical 3D and 2D interleaving models for double diffusive interleaving at the oceanic front (Stern, 1967; Toole and Georgi, 1981; McDougall, 1985a, 1985b; Niino, 1986; Yoshida et al., 1989; Kuzmina and Rodionov, 1992; May and Kelley, 1997; Kuzmina, 2000; Kuzmina and Zhurbas, 2000). Given that the intrusions in the oceanic fronts have different forms, the present results may be useful for interpreting empirical data. However, the simplicity of our model does not allow us to determine the maximum growth rate. Here again we can see an analogy with the work by Stern (1967).

Indeed, in a well-known paper by Stern (1967), which was the first study of the double diffusion instability of the infinite thermohaline front, the magnitude of the fastest growing mode was not found. The reason is that the growth rate in Stern's model could indefinitely increase with the horizontal wave number due to the neglect of vertical friction. A similar feature is typical of a simple model described above. The growth rate increases with the increase in the wave number $k$ up to the limit $\tilde{k}$ for which the constraint of $\tilde{k} \ll f/(2H_0 N_0)$ is still valid. Nevertheless, for a rough estimate of the time of formation of unstable perturbations it is reasonable to use formula (16). It is also worth evaluating the relationship between the growth rate of unstable disturbances and the layer thickness (case 1) or the characteristic vertical scale of disturbances (case 2). Let us address Eq. (17), which follows from the boundary conditions for one of the problems of studying the instability in a finite layer. The parameter $\chi = \mathrm{Re}(-aH_0^{\,2}) = 0.5(ks/K)^{1/2} H_0^{\,2}$ governs Eq. (17). The higher the value of this parameter, the greater the wave number of the unstable mode for the given parameters of the problem, $K$, $s$, and $H_0$, and therefore, the greater the growth rate. However, the applicability of our model imposes a constraint on the space of wave numbers, $k \ll f/(2N_0 H_0)$. In order to satisfy these two conditions simultaneously in the wide range of variability of hydrological parameters in the ocean, it is reasonable to put $1 \leq \chi \leq 2$. For $\chi = 2$, taking into account (16), we obtain the following formula relating the growth rate of disturbances and the vertical scale of the layer: $kc_2 = 1/T = 10 \cdot K/H_0^{\,2}$.

For damped with increasing $|z|$ solutions $F_3(z)$, the length scale $H = 2(K/ks)^{1/4}$ determines the characteristic vertical scale of the disturbances. Therefore, when $\chi = 2$ the characteristic vertical scale of disturbances is a scale on which the perturbation amplitude decreases by a factor $e = 2.718....$The formula relating the growth rate and the vertical scale of the disturbances will be the same as the previous one, but with $H$ instead of $H_0$.

It is easy to understand the physical meaning of the parameter $\chi$. This parameter characterizes the ratio of advection and vertical diffusion terms depending on the wave number $k$. Indeed, if we take into account that in our model $U = U_1 + U_3$ and take zero geostrophic velocity on the boundaries of the layer ($U_3 = -sH_0^{\,2}/2$, $s > 0$), the maximum velocity at the midline of



the layer will be $U_{max} = sH_0^2/2$. This allows the squared parameter $\chi$ to be presented as $\chi^2 = 0.25(ks/K)H_0^4 = 0.5 \cdot R_d kH_0$,

where $R_d = 0.5sH_0^3/K = U_{max}H_0/K$ is a diffusion analogue of the Reynolds number or Peclet number.

It is worth noting that Eq. (13), being differentiated, corresponds to a simplified form of the Orr–Sommerfeld equation (see e.g. (Lin, 1955; Stern, 1975)) written under the extra-long-wave approximation. However, there are a number of differences

between these equations, namely: (a) the destabilising factor in the Orr–Sommerfeld equation is friction, rather than diffusion, (b) the unknown function is a stream function in the vertical plane, and, finally, (c) to analyse instability in the frame of the Orr–Sommerfeld equation it is suggested that viscosity is vanishingly small but finite. For this reason, the disturbances out of the crirical layer are described by the equation of an ideal fluid, but in the region of a thin critical layer the equation of the so-called "viscous regime" is used (see, for example, Miles, 1965).

To obtain the solutions for the "viscous regime", the Orr–Sommerfeld equation is greatly simplified: only the terms describing derivatives of the unknown function of the 4th and 2nd order are considered (Iordanskiy and Kulikovskiy, 1966). Moreover, the velocity of the flow $U(z)$ is linearised as $U(z) \approx U'_z(z_0)(z - z_0)$, where $z_0$ is a point lying on the midline of the critical layer. In this regard, the solutions obtained in the present work for a parabolic type of geostrophic flow are different from the solutions of the "viscous regime" of the Orr–Sommerfeld equation, which are expressed by Hankel functions of order 1/3

(Lin, 1955). The unstable modes described by solutions (15) cannot be attributed to the instability of the critical layer: see formulas (16) describing the phase velocity of unstable modes. Thus, there are significant differences between the approaches used to study instability in the frame of the Orr–Sommerfeld equation [see also the paper by Miles (Miles, 1965)] and the approach proposed in the present work.

To conclude this section, we note the following.

The instability of the weak geostrophic flow in the frame of the solutions (15) is an oscillatory instability (the growth rate has real and imaginary components). Generally, using interleaving models (Stern, 1967; Tool and Georgy, 1981; McDougal, 1985a, 1985b; Niino, 1986; Yoshida et al., 1989; Kuzmina and Rodionov, 1992; May and Kelley, 1997; Kuzmina and Zhurbas, 2000; Walsh and Ruddick, 2000; Merryfield, 2002), it is possible to obtain the monotonous unstable modes only (the phase velocity of the disturbances is equal to zero: $\mathrm{Re}\, c = 0$.). The exceptions to this rule are the interleaving models describing the interleaving in

the equatorial fronts. In accordance with the modelling efforts (Richards, 1991; Edwards and Richards, 1999; Kuzmina et al., 2004; Kuzmina and Lee, 2005), the instability of the equatorial fronts in the scale of intrusive layering is regarded as an oscillatory instability.

The general solution (15) is one of the classes of solutions of Eq. (13). Thus, for example, at $ikc = 9a \cdot K + ikU_3$ it is possible to find an analytically general solution too. This solution will have a more complex structure than (15). Detailed analytical

consideration of unstable modes based on the analysis of different classes of solutions of Eq. (13) taking into account friction may be a subject for further research. To clearly define the range of applicability of our model, it is worth solving the eigenvalue problem for Eq. (7) for small values of parameter $\delta^2$ by means of numerical methods. This problem may be the subject of further research too. The analytical solutions found can be used to validate numerical solutions of the eigenvalue problems. Moreover, the analytical solutions obtained give analytical formulas for own functions, phase velocities and growth/decay rates

of disturbances that cannot, as a rule, be found exactly from numerical solutions.





## 3 Application to thermohaline intrusions in the Eurasian Basin of the Arctic Ocean

It is worth evaluating the time of formation of the large-scale intrusions based on the results of the presented model. According to Kuzmina et al. (2011), in the upper layer of the Polar Deep Water (PDW) where the large-scale intrusions are observed in the Eurasian Basin at stable–stable stratification, the following estimates of $f$ and $N$ are typical: $f = 1.4 \cdot 10^{-4}$ s$^{-1}$ and $N \approx 2 \cdot 10^{-3}$ s$^{-1}$. Therefore, for disturbances, for example, with the vertical scale $h = 100$ m the Rossby radius of deformation is only $hN/f \approx 1$ km.

According to the derivation of Eq. (7), the value of the linear shear $s$ is limited by the inequality of $|s|fL << N_0^2$. Given that the horizontal scale of the baroclinic fronts (along the cross-front axis $y$) in the upper layer of the PDW is approximately $L \approx 50$ –100 km [see examples of transections across the fronts of different types observed in the PDW (Kuzmina et al., 2011)], the maximal linear shear can be estimated as $|s| \leq (1–2)10^{-7}$ m$^{-1}$s$^{-1}$. Vertical diffusivity $K$ can be estimated in the range of $K = 10^{-6}$– $3 \cdot 10^{-6}$ m$^2$s$^{-1}$ (Merryfield, 2002) [see also the following paper where the evaluations of coefficients of diffusivity in the Arctic thermocline were considered (Walsh and Carmack, 2003)]. The typical vertical scale of intrusive layering in the fronts of PDW is approximately 30–40 m (Merryfield, 2002; Kuzmina et al., 2014). Let us evaluate the time formation of intrusions with the vertical scale ~ 40m. Using the following formula (see Subsection 2.3), $k_0 c = 10 \cdot K/H_0^2$, we can obtain that the time formation of the unstable mode is estimated as $1/(k_0 c_2)$ ~ 5 years at $K = 10^{-6}$ m$^2$s$^{-1}$ and approximately 2 years at $K = 3 \cdot 10^{-6}$ m$^2$s$^{-1}$.

To verify the applicability of our model it is necessary to estimate the wave number $k_0$, using the following formula (see Subsection 2.3):

$$k_0 = 16 \cdot K/(H_0^4 s). \tag{20}$$

Substituting $H_0 = 40$ m, $s = 2 \cdot 10^{-7}$ m$^{-1}$s$^{-1}$, $K = 10^{-6}$ m$^2$s$^{-1}$ in (20), we find $k_0 = 0.2 \cdot 10^{-3}$ m$^{-1}$. The value of $k_0 \approx 10^{-4}$ m$^{-1}$ may be obtained at $s = 2 \cdot 10^{-7}$ m$^{-1}$s$^{-1}$, $K = 3 \cdot 10^{-6}$ m$^2$s$^{-1}$. This value of $k_0$ lies in the wave number range of applicability of our model, since $\delta^2 \approx 4 \cdot 10^{-3} - 4 \cdot 10^{-4}$.

The above-presented estimates of the formation time of intrusions in PDW are evidently better than the evaluations that can be obtained from 2D modelling of baroclinic front instability (see Introduction).

In closing this section, let us justify the assumption that the circulations associated with changes in vorticity $\Delta p$ are not essential in the description of the formation of intrusions in all the cases considered. According to Eqs. (2) and (10), the characteristic scale of vertical velocity in such circulations can be written as $w_1 \sim U \cdot u \cdot H/(\tilde{L}^2 f)$. In all the above-considered cases of the application of the model to the Arctic intrusions, the following ratio is satisfied $U/\tilde{L} < 10^{-8}$ s$^{-1}$. Given that small disturbances of horizontal velocity cannot exceed the value of geostrophic velocity $U$, we find $w_1 < 4 \cdot 10^{-11}$ m s$^{-1}$. A liquid particle with such vertical velocity travels less than 0.004 m for the period of intrusion formation ($1/(k_0 c_2)$ ~ 3 years), while due to vertical diffusion, the fluid particle can cover a distance of 40 m in approximately the same period. Note also that damped with increasing $|z|$ solution $F_3(z)$ can be used for description of intrusions generation even if vertical velocity is not negligible quantity.



## 4 Conclusions

In this paper, we investigated analytically the 3D instability of a baroclinic front in quasi-geostrophic, long-wave approximation taking into account the vertical diffusion of buoyancy. It is shown for the first time that not only double diffusion, but the diffusion of buoyancy can cause destabilization of the geostrophic flow. Such instability has to be distinguished from the 2D McIntyre instability (McIntyre, 1970), a type of instability due to flow-dependent fluctuations in turbulent diffusivities (Smyth and Ruddick, 2010) and the 2D baroclinic instability due to double diffusion (Kuzmina and Rodionov, 1992; May and Kelley, 1997; Kuzmina and Zhurbas, 2000; Kuzmina and Lee, 2005).

In contrast to the work by Miles (Miles, 1965), in which it was shown that the influence of vertical diffusion of buoyancy is not essential in comparison with the influence of vorticity change to destabilize the zonal flow, we considered the opposite case, when vertical diffusion of buoyancy can play an important role as a destabilization factor of very weak geostrophic current with linear shear and large cross-frontal scale.

The model we developed can be considered as a modification of Stern's (Stern, 1967). However, instead of analysing the instability of a purely thermohaline front due to the double diffusion (Stern, 1967), in our case the instability of a weak baroclinic front is analysed taking into account the vertical diffusion of density. This model can be useful for describing stable and unstable disturbances of a planetary scale in the polar regions of the ocean under the stable–stable stratification, particularly in the deep water of the Arctic Basin, where weak baroclinic fronts with a large horizontal (cross-frontal) scale and typical temporal variability period of the order of several years or more are observed, and the beta-effect is negligible.

The stable (decaying with time) solutions are shown to describe long-wave disturbances that, unlike Rossby waves, can move not only to the west but also to the east, depending on the magnitude and sign of the linear shear of geostrophic velocity. It is important to underline that the linear shear of mean flow (parabolic dependence of mean velocity upon vertical co-ordinate) has an action upon the dynamics of disturbances and likewise the $\beta$-effect.

Unstable (increasing with time) solutions are used to describe the formation of large-scale intrusions in the areas of baroclinic fronts, which are observed in the Arctic Basin in the regions characterized by an absolutely stable stratification, for example, in the upper layer of the PDW in the Eurasian Basin.

The proposed description of intrusions generation in baroclinic fronts can be considered as a possible alternative mechanism relative to the differential mixing. However, at the moment this is just a hypothesis, and further efforts, both in theoretical modelling and field data analysis, are needed to justify it.

**Acknowledgements**

This work was supported by the Russian Science Foundation (Grant No 14-50-00095) and the Russian Foundation for Basic Research (Grant No 15-05-01479-a). The author is grateful to Victor Zhurbas for the constructive discussions of the results.

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



**Figures**

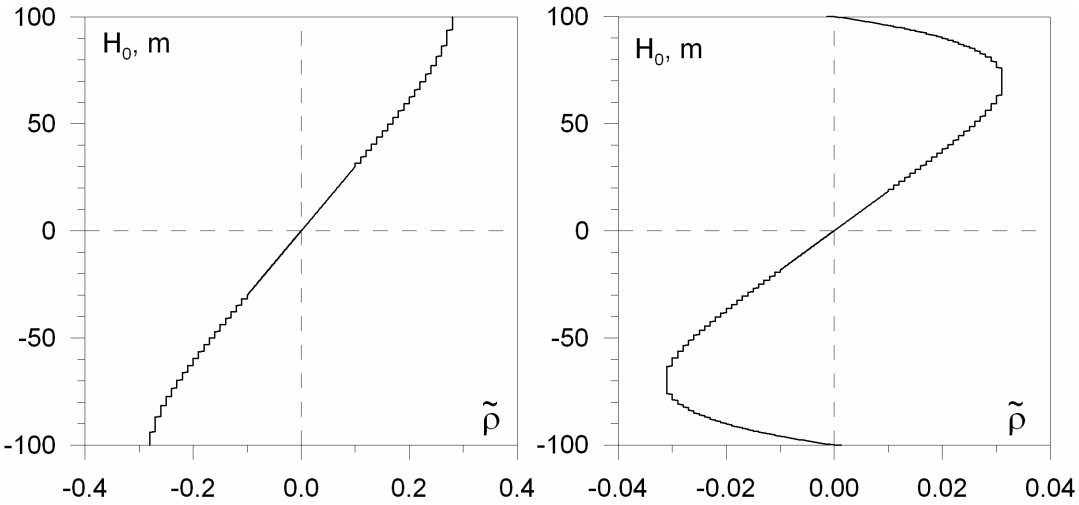

**Figure 1. Modelled vertical profiles of density disturbances** $\mathrm{Re}(dF/dz) = \mathrm{Re}\,\rho = \tilde{\rho}$ **for case 1. Unstable (growing) solution**

for boundary conditions of type 1 (left) and type 2 (right) and value of $\chi = \mathrm{Re}(-aH_0^{\,2}) = 0.5(ks/K)^{1/2}H_0^{\,2} = 1.5$, $K = 10^{-5}$

5    $\mathrm{m^2\,s^{-1}}$, $H_0 = 100$ **m,** $s = 10^{-7}$ **m$^{-1}$s$^{-1}$.**

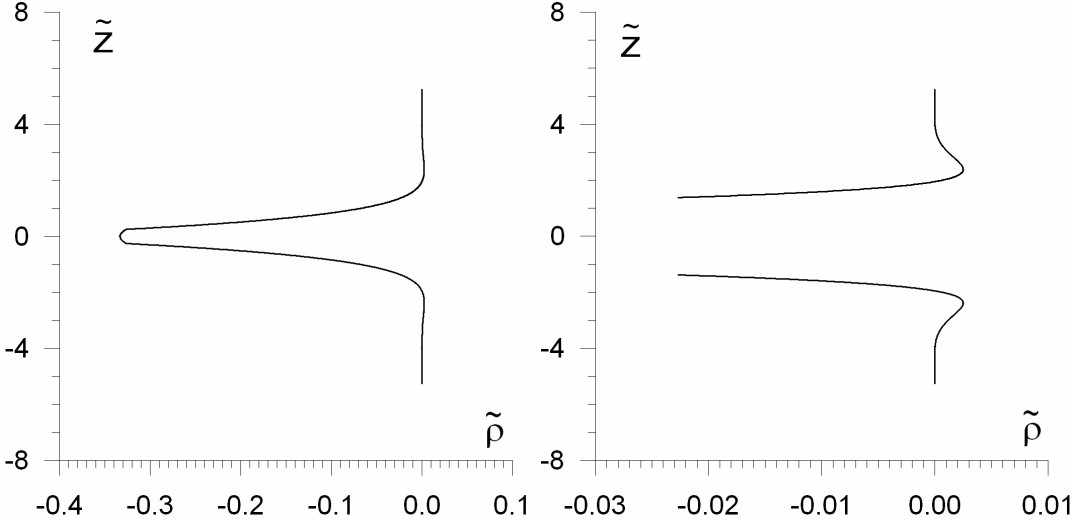

**Figure 2. Modelled vertical profile of density disturbances** $\mathrm{Re}(dF/dz) = \mathrm{Re}\,\rho = \tilde{\rho}$ **for case 2. The function** $\mathrm{Re}(dF_3/dz)$

**(left) and fragment of this function (right) depending on dimensionless co-ordinate** $\tilde{z} = z \cdot (ks/K)^{1/4}$ **are presented for**

$k = 10^{-5}$ **m$^{-1}$,** $s = 10^{-7}$ **m$^{-1}$s$^{-1}$,** $K = 10^{-5}$ **m$^2$ s$^{-1}$ .**




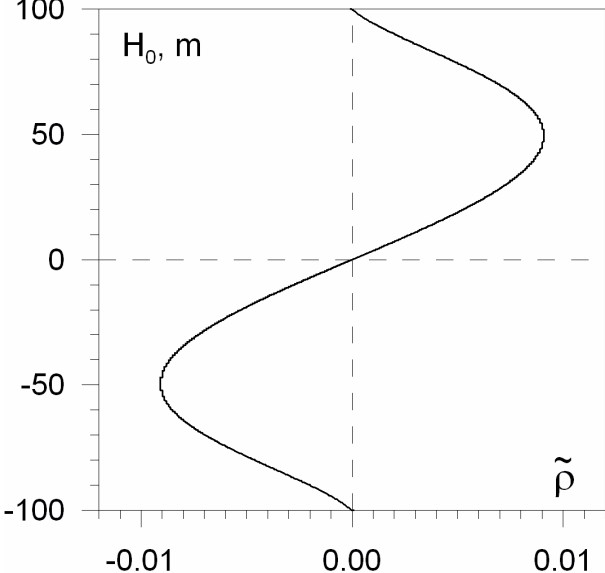

**Figure 3. Stable solution for case 1 and boundary conditions of type 2 at** $\chi = \mathrm{Re}(-aH_0^{\,2}) = 0.5(ks/K)^{1/2}H_0^{\,2} = 2$ , $K = 10^{-5}$ $\mathrm{m}^2\,\mathrm{s}^{-1}$ , $H_0 = 100$ **m,** $s = 10^{-7}$ **m⁻¹s⁻¹.**