# Peer review of "Generation of large-scale intrusions at baroclinic fronts: an analytical consideration with a reference to the Arctic Ocean"

_Ocean Science, 2016_

## Referee Comment (RC1) · Anonymous Referee #1 · 18 Aug 2016

While I am not overly enthusiastic about this paper, I believe that it would be suitable for publication after the grammar and sentence composition is improved.

---

## Referee Comment (RC2) · Anonymous Referee #2 · 22 Aug 2016

The author addresses an issue of destabilization of the geostrophic flow by vertical diffusion of density and speculates that such process might be the relevant in explaining the formation of intrusive layers in the Arctic Ocean. I enjoyed reading this paper and it is particularly pleasing to see the theoretical analysis. While I believe that there is a potentially great scientific merit to this work, I have several concerns about the clarity of the presentation, novelty of the theoretical results, and applicability to the Arctic Ocean. Please find my detailed comments and questions below that are aimed at improving the quality of the paper.

1. The title of the paper makes a clear reference to the Arctic Ocean, however the analysis of the instability presented here is not restricted to any specific part of the

[Figure]

ocean. In particular, a direct quantitative comparison of the theoretical results to the intrusions in the Arctic Ocean is not extensive and occupies only a small fraction of the manuscript. I would thus recommend focusing one topic: either on application to the Arctic intrusions or on a discovery of a new type of fluid dynamical instability.

2. The author claims that this manuscript shows for the first time that a diffusion can destabilize the geostrophic flow. However, the novelty of the presented findings can be questioned as a discussion of the relevant scientific literature on viscous instabilities is not present. In particular, the author should discuss the work of McIntire (1970), Baker (1970), and Calman (1976) who consider experimentally and analytically the diffusive instability. In addition, a discussion of Murno (2010) experiments, which show the importance of viscosity, needs to be present. These are just several of the papers that came to mind, I'm sure there is more literature on the visco-diffusive instabilities of geostrophic flows.

3. Writing out the QG equations with the stratification parameter N that depends on y is not common. There should be a reference to a book or a paper that presents its proper derivation i.e. an asymptotic expansion in Ro number where $N=N0+O(Ro)$. I'm not certain, but there might be some terms might be missing in Eq. 7 if $N=N(y)$ – please check and give a reference.

4. When neglecting the betta effect please provide quantitative estimates of a latitude at which betta-effect becomes less important (e.g. at 75 degree latitude betta is 25% of its value at the equator – is that beta negligible compared to shear term?). If betta-effect from your scaling end up being negligible at any latitudes then the instability that you consider should be of a small horizontal length scale.

5. A discussion of why mass and momentum diffusivities are assumed to be the same is missing. Note, that in a non-turbulent regime, which might be adequate for the deep Arctic, the viscosity is an order of magnitude larger than heat diffusivity and three orders larger than salt diffusivity.

6. Deriving Eq. 9 from Eq. 8 requires vertical integration since p_z=-g rho and I'm not sure what was done with the vertical integral of the term U*p_zzx when U=U(z). Perhaps showing more steps would clarify things.

7. Eqns. 2 and 10 should contain the physical mechanism behind the instability which was not explained throughout the paper. The authors take a dry math approach to the instability problem by calculating the growth rates; however, omitting the physical mechanism of the instability dramatically reduces the understanding of the problem for the readers. Perhaps a schematic showing a positive feedback loop would be helpful.

Also missing is a discussion of fundamental reasons for the existence of this instability i.e. does it release a potential energy of mean flow, does it feed on its kinetic energy or something else?

8. In assessing the stability properties of Eq. 11 the author assumes a special case of $U(z) \sim z^2$ and present analytical solutions (Eq 15). It is not clear to me how was the growth rates in Eq. 16a,b calculated from Eq. 15.

9. Eq. 11 that determines the stability of the flow does not have any y-derivatives and hence the stability properties do not depend on the y-direction wavelength. Thus, it looks like this instability is a 2D (in x,z–plane) rather than the 3D instability that the author claims to have investigated.

10. The authors demonstrate that Eq. 11 has unstable solutions for $U \sim z^2$ but it is not obvious that other, more realistic, profiles of U(z) can also lead to an instability. It would be useful to solve Eq. 11 via the eigenvalue decomposition in z and show that arbitrary profiles of U (that have curvature in z) are indeed unstable.

11. The meaning of discussion of limits at z=+-inf is not clear and needs to be organized better. In particular at z=+-inf U->inf which is unrealistic for the ocean and for the theory which assumes U to be small in a QG sense. Why not using finite domain size z-=[0 H] and a corresponding no flux boundary conditions?

12. It is not physical to have a growth rate that grows with increasing wavenumber as it implies that any kind of small scale noise would be preferentially amplified. The author motivates the paper with the idea that the size of intrusions in the arctic ocean might be explained by an instability. However, there seems to be no preferential wavelength at which the instability occurs and hence one cannot expect the appearance of intrusions of particular height.

In addition, the theory breaks down at a particular length scale which the authors choose as the scale of intrusions and use it to calculate the growth rates. It is questionable to use these estimates since the theory technically does not apply at this marginal scales (i.e. the neglected terms need to be included).

13. Because there is no high wavenumber cutoff it is questionable whether the numerical model results shown in Fig. 1 are realistic; the step formation shown in Fig. 1a might be at the size of the numerical grid and hence their dynamics is not adequately resolved.

14. A discussion of Orr-Sommerfeld equations seems unnecessary as it only makes a mathematical connection with insufficient improvement of our physical understanding of the problem; thus, it only makes the paper harder to understand.

15. Application to the Arctic Ocean can be questioned because i) there is no preferential length of instability that can be compared with the size of intrusions and ii) the growth rates are of the order of years are too large because the mean currents will most likely significantly change on the long time and very small spatial scales of the instability.

16. I'd suggest working on the brushing up the grammar and logical presentation of the paper. Many paragraphs do not contribute well to the clarity of the paper and can be outright deleted. The title can be clearer as well: e.g. Generation of large-scale intrusions via diffusive instabilities.

References:

Baker, D. James. "Density gradients in a rotating stratified fluid: experimental evidence for a new instability." Science 172.3987 (1971): 1029-1031.

Calman, Jack. "Experiments on high Richardson number instability of a rotating stratified shear flow." Dynamics of Atmospheres and Oceans 1.4 (1977): 277-297.

Munro, R. J., M. R. Foster, and P. A. Davies. "Instabilities in the spin-up of a rotating, stratified fluid." Physics of Fluids (1994-present) 22.5 (2010): 054108.

McIntyre, Michael E. "Diffusive destabilisation of the baroclinic circular vortex." Geophysical and Astrophysical Fluid Dynamics 1.1-2 (1970): 19-57.

---

## Author Comment (AC1) · 30 Aug 2016

I am very grateful to you for the review. I really appreciate your opinion on my work. In order to improve your opinion about this work I would like to emphasize the following. In this paper I propose a new approach to describe intrusions in the Arctic region. The solutions found have not been known before. (Finding the exact solution of a relatively complicated equation may be referred as a lucky case but it requires a lot of work.) It seems to me that the solutions found will contribute to deeper understanding of intrusive layering in the Arctic at stable-stable stratification.

I apologize for the grammar and syntax errors - the necessary efforts will be made to improve the text.

---

## Author Comment (AC2) · 30 Aug 2016

I am pleased that this article has caused you concern. I am very grateful to you for your comments. The comments and questions are extremely useful for me. I will try to take them into account during the article revision.

---

## Author Response (AR1)

Reviewer #1

*While I am not overly enthusiastic about this paper, I believe that it would be suitable for publication after the grammar and sentence composition is improved.*

Great efforts have been made to improve the grammar and sentence composition (see the makeup_revision (os-2016-15-supplement-version1.pdf)).

Reviewer #2

*1. The title of the paper makes a clear reference to the Arctic Ocean, however the analysis of the instability presented here is not restricted to any specific part of the ocean. In particular, a direct quantitative comparison of the theoretical results to the intrusions in the Arctic Ocean is not extensive and occupies only a small fraction of the manuscript. I would thus recommend focusing one topic: either on application to the Arctic intrusions or on a discovery of a new type of fluid dynamical instability.*

In view of this remark, we changed the title for "Generation of large-scale intrusions at baroclinic fronts: an analytical consideration with a reference to the Arctic Ocean". The new title is focused on the analysis of the instability and, nevertheless, contains a reference to the Arctic Ocean. To my mind the reference allows to emphasize that (a) this study was primarily aimed to understand the nature of large-scale intrusions observed in the Arctic Ocean, and the aim was directly stated in the Introduction, (b) the approximations and simplifications used in the model, such as the weak geostrophic currents, the wide baroclinic fronts, the small Burger number, the small parameter beta, were fitted to the Arctic Ocean conditions (see Section 2), and (c) the theoretical results were compared with the observations of intrusions in the Arctic Ocean (see Section 3).

*2. The author claims that this manuscript shows for the first time that a diffusion can destabilize the geostrophic flow. However, the novelty of the presented findings can be questioned as a discussion of the relevant scientific literature on viscous instabilities is not present. In particular, the author should discuss the work of McIntire (1970), Baker (1970), and Calman (1976) who consider experimentally and analytically the diffusive instability. In addition, a discussion of Murno (2010) experiments, which show the importance of viscosity, needs to be present. These are just several of the papers that came to mind, I'm sure there is more literature on the visco-diffusive instabilities of geostrophic flows.*

You are absolutely right, drawing attention to a very important and delicate things. For clarity of presentation of the work more precise explanations are given now (p.8, lines 9-20; p.11, lines 19-27 in os-2016-15-manuscript-version3.pdf). First of all, I added to the manuscript an explanation of the differences between the McIntyre instability and instability obtained in this work (p.8, lines 9-20). For convenience, I reproduce some of the explanations here.

In accordance with (11) that was obtained from Eqs (1)-(10) at Bu<<1 (or at Ri>>1) and Pr=1, the large-scale disturbances can be unstable. Such instability has to be distinguished from the diffusive instability (McIntyre, 1970; Baker, 1970; Calman, 1976). Indeed, instability in the model by McIntyre occurs when Ri <(Pr + 1) ** 2 / 4Pr and is absent at Pr = 1.

The articles mentioned in your list either confirmed experimentally the McIntyre instability (Baker, 1970; Calman, 1976; Murno (2010)) or contained some useful additions to the McIntyre's theory (Calman, 1976). All of the mentioned articles were added to the References.

One of the important distinctions between these two models of baroclinic front instability is that in the present model the disturbances are allowed to have a nonzero slope in the along-front direction while in the model of diffusive instability by McIntyre (1970) the slope is taken zero.

Therefore, the McIntyre's model and other models in which the term $\partial p / \partial x \, \partial p / \partial x$ in the equations of motions is ignored (McIntyre, 1970; Calman, 1996; and others) can be referred as the 2D models (See also below the response to Comment No 9). Sometimes such models are called the models of symmetric instability.

From the mathematical point of view, the models that take into account the along-front slope of the perturbations, are much more complicated. Indeed, the analysis of the instability in the 2D models ultimately reduces to finding the roots of a polynomial depending upon the wave-number and growth rate. The models, that take into account the along-front slope of the perturbations, are reduced to the differential equations with variable coefficients, and such problems can be solved analytically only in rare cases.

*3. Writing out the QG equations with the stratification parameter N that depends on y is not common. There should be a reference to a book or a paper that presents its proper derivation i.e. an asymptotic expansion in Ro number where N=N0+O(Ro). I'm not certain, but there might be some terms might be missing in Eq. 7 if N=N(y) – please check and give a reference.*

It was said on p.3, lines 18-19 and 27 of os-2016-15-manuscript-version2.pdf that the dependence of the Brunt-Vaisala frequency upon the coordinate y is weak, that is, the inequality |s|fL << N0 ** 2 is satisfied. For this reason the additional terms in Eq. (7) can be neglected (it was pointed out on p. 3, line 27 of os-2016-15-manuscript-version2.pdf). A more detailed explanation of the issue is done in the revised manuscript (p.3, lines 25-26 of os-2016-15-manuscript-version3.pdf).

*4. When neglecting the betta effect please provide quantitative estimates of a latitude at which betta-effect becomes less important (e.g. at 75 degree latitude beta is 25% of its value at the equator – is that beta negligible compared to shear term?). If betta effect from your scaling end up being negligible at any latitudes then the instability that you consider should be of a small horizontal length scale.*

The value of parameter beta in the vicinity of intrusions observation is given on p. 10, lines 21-22 of the revised MS (os-2016-15-manuscript-version3.pdf). The comparison of the beta and shear terms is done on p.10, lines 27-28 of os-2016-15-manuscript-version3.pdf.

Of course, the larger the beta (i.e. the lower the latitude) the smaller horizontal length scale disturbances remain unaffected by the beta-effect. However, in the Arctic Ocean, near the Pole, beta is small, N/f~10, Bu<<1 for large-scale disturbances. The beta-effect term in Eq.(7) has approximately the same order of smallness as the relative vorticity, which is neglected in our model: Bu*U*k~beta*Bu/k. For this reason we can consider the large-scale (50-100 km) disturbances near the Pole.

In Section 3 of revised MS the values of all parameters are presented to confirm the correctness of our approach.

*5. A discussion of why mass and momentum diffusivities are assumed to be the same is missing. Note, that in a non-turbulent regime, which might be adequate for the deep Arctic, the viscosity is an order of magnitude larger than heat diffusivity and three orders larger than salt diffusivity.*

If the Eurasian Basin of the Arctic in the depth range of 600–1200 m is characterized by the non-turbulent regime, the existing models of interleaving will forecast the unstable modes with very small vertical length scale, which is obviously contrary to the observations. Merryfield (2002) suggested that this depth range is characterized by an intermittent turbulence and introduced a notion of differential mixing to parameterize the vertical diffusion terms. As a result, a satisfactory agreement between the vertical length scale of unstable modes and the thickness of

observed intrusions in a purely thermohaline front has been achieved (a 3D model of the thermohaline front instability).

When the differential mixing parameterization was applied to the 2D model of the baroclinic front instability, a large difference was found between vertical scales of the unstable disturbances and the observed intrusions. A comprehensive discussion on the issue was presented in Kuzmina et al. (2014). If a suggestion on non-turbulent diffusivities is used in the 2D model of the baroclinic front instability, all unstable modes, including the maximum-growing one, will be of no more than several centimeters vertical scale.

This study suggests a weak turbulent rather than molecular regime in the deep Arctic layer under consideration, i.e. the difference between the momentum and mass exchange coefficients exists, but it is not high, obeying the condition of $Pr*Bu<<1$.

Much of what is said here, was presented in the previously submitted manuscript, but I still made some additions/explanations to the revised manuscript (p.10, lines 29-30 of os-2016-15-manuscript-version3.pdf).

*6. Deriving Eq. 9 from Eq. 8 requires vertical integration since p_z=-g rho and I'm not sure what was done with the vertical integral of the term U*p_zzx when U=U(z). Perhaps showing more steps would clarify things.*

The easiest way to show that Eq. 9 was derived from Eq. 8 correctly, is by differentiating Eq. 9 with respect to z for W0 = const (see the explanation on p. 4, lines 16-17 of os-2016-15-manuscript-version3.pdf).

Also, you can carry out the integration procedure of Eq. 8, applying the rule of integration by parts as follows:

$$\int \left( \frac{\partial}{\partial t} + U \frac{\partial}{\partial x} \right) \left( \frac{\partial^2 p}{\partial z^2} \right) dz - \int vsf dz = K \frac{\partial^3}{\partial z^3} p + Const =$$

$$\frac{\partial}{\partial t} \partial p / \partial z + \int U \frac{\partial}{\partial x} \frac{\partial^2 p}{\partial z^2} dz - \int \frac{\partial p}{\partial x} s dz = K \frac{\partial^3}{\partial z^3} p + Const \ . \qquad (8^*)$$

Here we took into account that $fv = \partial p / \partial x$.

Second term in the left of Eq. $(8^*)$ we can be rewritten using the rule of integration by parts as follows:

$$\int U \frac{\partial}{\partial x} \frac{\partial^2 p}{\partial z^2} dz = \frac{\partial}{\partial x} U \frac{\partial p}{\partial z} - \frac{\partial}{\partial x} psz + \int \frac{\partial p}{\partial x} s dz$$

Here we took into account that $U = sz^2 / 2$. Substituting the latter into Eq. $(8^*)$ we can obtain Eq. (9), remembering that $\partial p / \partial z = -g\rho$ and $v\partial\overline{\rho} / \partial y = vfsz / g$ .

I doubt the need to include in the article as a detailed description of the integration procedure – I will do that if the Reviewer and the Editor consider it useful.

*7. Eqns. 2 and 10 should contain the physical mechanism behind the instability which was not explained throughout the paper. The authors take a dry math approach to the instability problem by calculating the growth rates; however, omitting the physical mechanism of the instability dramatically reduces the understanding of the problem for the readers. Perhaps a schematic showing a positive feedback loop would be helpful. Also missing is a discussion of fundamental reasons for the existence of this instability. i.e. does it release a potential energy of mean flow, does it feed on its kinetic energy or something else?*

In some cases, it is very difficult to elaborate the physical mechanism behind the instability. It is especially difficult in the case of the oscillatory instability. Remember the case of the instability of the critical layer: physicists could not explain it for a long time (only in 1975 Stern (1975) presented an interesting simple explanation of the critical layer instability).

With regard to our case of instability, it is possible to propose the following physical reasoning. As we can see from Eq. (16), the phase velocity of unstable disturbances is directed along the geostrophic current and exceeds the maximum velocity of the current. In such a case, in my opinion, the most likely is the conversion the kinetic energy of the main flow into the kinetic energy of disturbances.

The above-presented physical reasoning was included to the revised MS (p.8, lines 21-26 of os-2016-15-manuscript-version3.pdf).

*8. In assessing the stability properties of Eq. 11 the author assumes a special case of U(z)_z^2 and present analytical solutions (Eq 15). It is not clear to me how was the growth rates in Eq. 16a,b calculated from Eq. 15.*

It is explained in the revised MS (see p.7, line 2 of os-2016-15-manuscript-version3.pdf).

*9. Eq. 11 that determines the stability of the flow does not have any y-derivatives and hence the stability properties do not depend on the y-direction wavelength. Thus, it looks like this instability is a 2D (in x,z–plane) rather than the 3D instability that the author claims to have investigated.*

Eq. (7) is a 3D equation, and Eq. (11) which has no y derivatives was derived from Eq. (7) at Bu<<1. The absence of the y-derivative in Eq. (11) does not mean that the pressure disturbance does not depend on y because the velocity disturbance u is determined by Eq. (2). Moreover, we consider a finite width front with the length scale L along latitude (i.e. along the y co-ordinate). On the lateral boundaries of the front (y=0, L) the following conditions are met: v(y=0)=0, v(y=L)=0 (in accordance with Eq. (7)). For this reason, we have to seek the solution of Eq. (11) in the form (12): all decision variables depend of 3 co-ordinates x, y, and z. Thus, in my opinion, our model is a special (simplified) case of the 3D model describing the extra-long disturbances.

Some explanations were added to the revised MS in view of this remark (see p. 6, lines 3-5 of os-2016-15-manuscript-version3.pdf). Nevertheless, in order to avoid the 3D-2D confusion, in Abstract and Conclusions of the revised MS, the model is no longer directly referred as the 3D model.

*10. The authors demonstrate that Eq. 11 has unstable solutions for U_z^2 but it is not obvious that other, more realistic, profiles of U(z) can also lead to an instability. It would be useful to solve Eq. 11 via the eigenvalue decomposition in z and show that arbitrary profiles of U (that have curvature in z) are indeed unstable.*

To my mind, it would not be worth to consider here one more problem, which is much more complicated than the present one. The most interesting case is to consider the main flow velocity in the form of U = U1 + U2 + U3. Such problem is beyond the scope of this paper, and will be considered in the future. Here we draw attention to the importance of considering the parabolic shape of the main flow. Note that the effect of linear shear on the perturbation dynamics has not been analytically studied before.

*11. The meaning of discussion of limits at z=+-inf is not clear and needs to be organized better. In particular at z=+-inf U->inf which is unrealistic for the ocean and for the theory which assumes U to be small in a QG sense. Why not using finite domain size z-=[0 H] and a corresponding no flux boundary conditions?*

For the analysis of differential equations with variable coefficients it is necessary to use standard mathematical methods, which have been presented in this paper. However, to construct the solutions that are useful for applications it is needed to consider a layer of finite thickness - it was stated in the previous version of manuscript (see p.5, lines 22-31; p.7, lines 8-31 of os-2016-15-manuscript-version2.pdf). In the revised MS, the additional explanations are presented (see p.7, lines 9-24, 29-30; p.8, lines 1-6 of os-2016-15-manuscript-version3.pdf).

*12. It is not physical to have a growth rate that grows with increasing wavenumber as it implies that any kind of small scale noise would be preferentially amplified. The author motivates the paper with the idea that the size of intrusions in the arctic ocean might be explained by an instability. However, there seems to be no preferential wavelength at which the instability occurs and hence one cannot expect the appearance of intrusions of particular height. In addition, the theory breaks down at a particular length scale which the authors choose as the scale of intrusions and use it to calculate the growth rates. It is questionable to use these estimates since the theory technically does not apply at this marginal scales (i.e. the neglected terms need to be included).*

The disadvantage of the model is its inability to forecast the characterictics of the most unstable mode – in the manuscript this issue has been discussed in detail (see p. 9, lines 11-17 of os-2016-15-manuscript-version2.pdf). However, it seems normal that the primary aim of the new instability problems is the proof of the potential for instability. For example, the pioneering work by Stern (1967), the first model of the DD interleaving, did not contain any estimate of the fastest growing mode because, like the present model, the growth rate increased unlimitedly with the wavenumber (it was discussed in os-2016-15-manuscript-version2.pdf too – see p.9, lines 11-17). On the other hand, in the studies of geophysical flow instabilities, some methods are used (e.g. the Rayleigh method) that can provide some conclusion about the possibility of instability, but cannot give the form of the unstable solutions and characteristics of the most unstable mode (remember e.g. the well-known work by J. Pedlosky(1964)).

The characteristics of the most unstable mode can be obtained by means of numerical integration of Eq. (7). Such a study being outside the scope of this paper is under way. However, the analytical considerations presented here showed that the unstable modes (not the fastest growing ones!) can occur at relatively large vertical wavelength of several tens of meters offering the principal possibility for explanation of the large-scale intrusions. Note that in all previous models of the baroclinic front interleaving, all unstable modes have had much smaller vertical wavelength.

*13. Because there is no high wavenumber cutoff it is questionable whether the numerical model results shown in Fig. 1 are realistic; the step formation shown in Fig. 1a might be at the size of the numerical grid and hence their dynamics is not adequately resolved.*

I'm sorry for the confusion - the small steps seen in Figs. 1 and 3 are an artefact caused by bad choice of the output data format used to store the results of numerical calculations. In the revised manuscript this annoying drawback has been corrected (see new Figs. 1 and 3).

*14. A discussion of Orr-Sommerfeld equations seems unnecessary as it only makes a mathematical connection with insufficient improvement of our physical understanding of the problem; thus, it only makes the paper harder to understand.*

I cannot withdraw fully the mentioning of the Orr-Sommerfeld equation because of the need to acquaint the reader with the important work by Miles (Miles, 1965).

Moreover, the analogies are often useful and can contribute to understanding the physics of the processes. Eq. (11) is a model (partial case) for the Orr-Sommerfeld equation. Also it is

necessary to underline that the growing with time solutions are not relevant to the critical layer instability.

However, in accordance with this comment I cut the paragraphs devoted to the discussion of the O-S equation in the revised MS (see p.11, lines 23-35; p.12, lines 1-3 of the makeup revision (os-2016-15-supplement-version1.pdf)).

There are only a few sentences about the critical layer instability left in the revised MS (see p.8, lines 21-24 of os-2016-15-manuscript-version3.pdf).

*15. Application to the Arctic Ocean can be questioned because i) there is no preferential length of instability that can be compared with the size of intrusions and ii) the growth rates are of the order of years are too large because the mean currents will most likely significantly change on the long time and very small spatial scales of the instability.*

Strictly speaking, there are no models of baroclinic front instability that could fully describe the formation of large-scale intrusions in the Arctic Ocean at the stable-stable stratification (see Introduction where the issue is discussed in detail). In other words, the use of all existing theories can be questioned.

It is worth remembering that in this paper only a hypothesis on possible mechanisms of the large-scale intrusions generation in the Arctic Ocean is suggested.

It is important, that the new modes of instability have vertical scale that can reach tens of meters. However, the model is so complex that much more efforts are needed to obtain the exhaustive results of modelling which may be fully comparable with the empirical data. This paper is just the first step of the studies.

As to the growth time estimate of the order of years – I do not think it is too large. Contrary, it is in accordance with the results by Merryfield (2000, 2002) who resulted in the estimate of the time of formation of intrusions and the time scale of variability of the mean currents in the deep Arctic Ocean as several years.

*16. I'd suggest working on the brushing up the grammar and logical presentation of the paper. Many paragraphs do not contribute well to the clarity of the paper and can be outright deleted. The title can be clearer as well: e.g. Generation of large-scale intrusions via diffusive instabilities.*

Great efforts have been made to improve the grammar and logical presentation (see the makeup revision (os-2016-15-supplement-version1.pdf)).

Some paragraphs were outright deleted (see p.9, lines 10-21; p.10, lines 25-29; p.11, lines 13-16; p.11, lines 23-35; p.12, lines 1-3 of the makeup revision (os-2016-15-supplement-version1.pdf)). The title has been changed.

[revised manuscript text omitted]